# Enhanced Growth Inhibition and Apoptosis Induction in Human Colon Carcinoma HT-29 Cells of Soluble Longan Polysaccharides with a Covalent Chemical Selenylation

**DOI:** 10.3390/nu14091710

**Published:** 2022-04-20

**Authors:** Ya-Hui Yu, Zhi-Mei Tang, Cen Xiong, Fei-Fei Wu, Jun-Ren Zhao, Xin-Huai Zhao

**Affiliations:** 1Key Laboratory of Dairy Science, Ministry of Education, Northeast Agricultural University, Harbin 150030, China; b20100128@neau.edu.cn; 2School of Biology and Food Engineering, Guangdong University of Petrochemical Technology, Maoming 525000, China; tzm1103@gdupt.edu.cn (Z.-M.T.); xiongcen5891@gdupt.edu.cn (C.X.); wuff@gdupt.edu.cn (F.-F.W.); zhaojunren@gdupt.edu.cn (J.-R.Z.); 3Research Centre of Food Nutrition and Human Healthcare, Guangdong University of Petrochemical Technology, Maoming 525000, China; 4Maoming Branch, Guangdong Laboratory for Lingnan Modern Agriculture, Guangdong University of Petrochemical Technology, Maoming 525000, China

**Keywords:** longan polysaccharides, selenylation, HT-29 cells, anti-cancer activity, apoptosis

## Abstract

The selenylated polysaccharides chemically belong to the organic Se-conjugated macromolecules and have recently been attracting more and more attention due to their potential to promote body health or prevent cancers. Longan (*Dimocarpus longan* L.), as a subtropical fruit, contains soluble and non-digestible polysaccharides that are regarded with health care functions in the body. In this study, the longan polysaccharides (LP) were obtained via enzyme-assisted water extraction, and then chemically selenylated using a reaction system composed of HNO_3_–Na_2_SeO_3_ to yield two selenylated products, namely, SeLP1 and SeLP2, with Se contents of 1.46 and 4.79 g/kg, respectively. The anti-cancer effects of the three polysaccharide samples (LP, SeLP1, and SeLP2) were thus investigated using the human colon cancer HT-29 cells as the cell model. The results showed that SeLP1 and SeLP2 were more able than LP to inhibit cell growth, alter cell morphology, cause mitochondrial membrane potential loss, increase intracellular reactive oxygen and [Ca^2+^]_i_ levels, and induce apoptosis via regulating the eight apoptosis-related genes and proteins including Bax, caspases-3/-8/-9, CHOP, cytochrome c, DR5, and Bcl-2. It was thereby proven that the selenylated polysaccharides could induce cell apoptosis via activating the death receptor, mitochondrial-dependent, and ER stress pathways. Collectively, both SeLP1 and SeLP2 showed higher activities than LP in HT-29 cells, while SeLP2 was consistently more active than SeLP1 in exerting these assessed anti-cancer effects on the cells. In conclusion, this chemical selenylation covalently introduced Se into the polysaccharide molecules and caused an enhancement in their anti-cancer functions in the cells, while higher selenylation extent was beneficial to the activity enhancement of the selenylated products.

## 1. Introduction

Selenium (Se) is an essential trace element to the body and is well-known for its biological effects on body growth and developments, especially for its ability to improve the activity of seleno-enzymes such as thioredoxin reductase and glutathione peroxidase, or its capacity to prevent free radicals from damaging cells and tissues [1]. However, insufficient dietary intake of Se is associated with serious events including gastrointestinal and prostate cancer, cardiovascular disease, diabetes, Keshan disease, and inflammatory disorders [2]. In most regions of the world, Se content in the soil and hence dietary Se intake are low [3]. Thus, Se supplements, especially Se-enriched food supplements, are beneficial for people living in these regions. Higher bioavailability of Se is mainly related to the organic Se because more than 80% of the organic Se can be absorbed in the body [4]. In contrast, the inorganic Se is poor in bioavailability. Fortunately, the inorganic Se can be transformed into the organic Se via covalent binding with proteins and polysaccharides by both biological and chemical approaches, resulting in higher bioavailability and safety [3,5]. More importantly, the Se-containing compounds were found able to inhibit tumor formation and cancer cell proliferation, while accumulating evidence also indicates that Se fortification in diets could reduce cancer risk [6,7,8]. It was evidenced in a nude mouse model injected with the wild-type colon cancer HT-29 cells that a Se-containing compound 4-phenylbutylisoselenocyanate could inhibit colon tumor growth [9], while another Se-containing compound, methylseleninic acid, could induce cell apoptosis and inhibit the growth of colon cancer HT-29 cells [10]. Se-enriched food supplements are thus regarded to have anti-cancer functions in the body.

Selenylated polysaccharides including the natural Se-containing polysaccharide or their synthesized derivatives are the organic Se-conjugated biomacromolecules, and have stronger bioactivities in comparison with the Se-free polysaccharides and inorganic Se such as anti-tumor, anti-oxidant, immune regulation, and anti-bacterial effects [3,11]. After a Se biotransformation, the fermented sourdough had increased bioaccessible organic Se, while the resultant hydrolysates could counteract the induced oxidative cell damage [12]. It was reported that the Se-containing polysaccharides from tea showed an anti-tumor effect on the mice carrying the S180 tumor [13], while the selenylated polysaccharides from *Eriobotrya japonica* and *Artemisia sphaerocephala* could induce cell apoptosis in human lung cancer (A549) and hepatocellular carcinoma (HepG2) cells [14,15]. In addition, the Se-containing polysaccharides from *P. fortuneana* were able to induce cell apoptosis and inhibit the growth of triple negative breast cancer MDA-MB-231 cells [16], while the selenylated pectic polysaccharides had obvious growth inhibition on the three cancer cells, namely, A549, BGC-823, and HeLa cells [17]. Furthermore, the selenylated *Grifola frondosa* polysaccharides had a stronger anti-cancer effect on the tumor-bearing mice [18]. Thus, the discovery of Se-containing polysaccharides with beneficial potentials to prevent cancer (e.g., colorectal cancer, CRC) has become a hotspot in the field of natural polysaccharides because CRC is the most common type of malignant tumor, with higher mortality and morbidity. Based on the 2020 Global Tumor Burden Report, CRC is one of the three cancers with the newest cases in the world, and has the second highest mortality proportion [19].

Longan (*Dimocarpus longan* L.) is an important subtropical fruit with several phytochemicals including polysaccharides and polyphenols, which have health benefits such as anti-cancer, memory enhancing, and anti-oxidant activities in the body [20,21]. The soluble longan polysaccharides (LP) were found to contain arabinose, mannose, glucose, and galactose, while the main linkages of the saccharide residues were (1^®^4)-β-glucose and (1^®^6)-β-mannose [22]. It was reported that LP had immunological activity [22], and could also inhibit the growth of two tumor cells, namely SKOV3 and HO8910 cells [23]. LP are thus considered as the main bioactive components that made a contribution to these health benefits of longan [24]. A chemical modification of polysaccharide can change polysaccharide structures. It is reasonable to assume that an application of chemical selenylation on LP might lead to enhanced activity of the selenylated LP (SeLP) such as anti-cancer, anti-oxidant, and immune potentials, which have still not been clarified thus far. Thus, such a study deserves our consideration.

In this study, the targeted soluble but non-digestible LP were extracted from the dried longan fruits using an enzyme-assisted water extraction, and then chemically selenylated into two selenylation extents using a reaction system composed of HNO_3_ and Na_2_SeO_3_ to prepare two respective SeLP products, namely SeLP1 and SeLP2. Afterward, the anti-cancer effects of SeLP1 and SeLP2 on human colon carcinoma HT-29 cells were explored using the unmodified LP as a control. The aim of this study was to reveal whether the performed chemical selenylation of LP might cause a change in its anti-cancer activities such as growth inhibition and apoptosis induction, especially whether the selenylation extent could govern the anti-cancer activity of SeLP to the cells.

## 2. Materials and Methods

### 2.1. Regents and Materials

The dried longan fruits were bought from Maoming Market, Guangdong Province, China. The human colon cancer cell line (HT-29) used in this study was purchased from the Cell Bank of Shanghai Institute of Biochemistry and Cell Biology (Shanghai, China).

Fetal bovine serum (FBS) was purchased from Wisent Inc. (Montreal, QC, Canada), while McCoy’s 5A medium and methyl thiazoly tetrazolium (MTT) were acquired from Sigma-Aldrich Co. Ltd. (St. Louis, MO, USA). 5-Fluorouracil (5-FU) and pectinase (30 kU/g) were obtained from Aladdin Bio-Chem Technology Co. Ltd. (Shanghai, China). Cellulase (50 kU/g) and phosphate-buffered saline (PBS) were purchased from Solarbio Science and Technology Co. Ltd. (Beijing, China), while the BCA Protein Assay Kit, Fura-2 pentakis (acetoxymethyl) ester (Fura-2 Am), dimethyl sulfoxide (DMSO), Annexin V-FITC apoptosis detection kit, Hoechst 33258, caspase-3 inhibitor Ac-DEVD-CHO, ROS assay kit, crystal violet staining solution, and mitochondrial membrane potential assay kit with JC-1 were purchased from Beyotime Institute of Biotechnology (Shanghai, China). RNAprep Pure Cell Bacteria Kit was purchased from Tiangen Biotech Co. Ltd. (Beijing, China), while the NovoScript^®^ Two-Step RT-PCR Kit and SYBR qPCR SuperMix Plus were purchased from Novoprotein Biotech Co. Ltd. (Suzhou, China). All chemicals used were of analytical grade, while water used was generated from the Milli-Q Plus system (Millipore Corp., New York, NY, USA).

Primary antibody β-actin (bs-0061R) was provided by Bioss Biotechnology Co. Ltd. (Beijing, China), while other primary antibodies including Bax (5023S), Bcl-2 (15071S), caspase-3 (9662S), caspase-8 (9746S), caspase-9 (9508S), CHOP (2895S), cytochrome c (11940S), and DR5 (69400S) were provided by Cell Signaling Technology (Shanghai) Biological Reagents Co. Ltd. (Shanghai, China). Additionally, the secondary antibodies horseradish peroxidase-labeled goat anti-rabbit IgG (ZB-2301) and goat anti-mouse IgG (ZB-2305) were provided by Zhongshan Golden Bridge Bio-Technology (Beijing, China).

### 2.2. Extraction and Selenylation of Longan Polysaccharides

The dried longan was mashed and mixed with distilled water to extract LP at a fixed solid–liquid ratio of 1:15 (w/v), as previously described [25], added with the respective pectinase and cellulase at 150 U and 300 U per gram dried longan as a past study did [26], adjusted to pH 4.5 by 1.0 mol/L HCl, and then kept at 50 °C for 4 h with gentle stirring. After centrifugation at 8000× *g* for 10 min, the separated supernatant was concentrated into one-twentieth volume using a vacuum rotary evaporator at 50 °C, and mixed with absolute ethanol of three-fold volume at 4 °C for 24 h. The collected precipitates were washed with absolute ethanol three times, and finally lyophilized to obtain LP.

The selenylated LP was prepared using the HNO_3_–Na_2_SeO_3_ method as previously described [27]. In brief, LP of 500 mg was dispersed in 0.5% HNO_3_ of 10 mL, mixed with 25 or 75 mg of Na_2_SeO_3_, and kept at 75 °C for 8 h with gentle stirring. After the reaction, the whole reaction system was centrifuged at 8000× *g* for 10 min, while the yielded supernatant was mixed with absolute ethanol of three-fold volume at 4 °C for 24 h. The collected precipitates were washed with anhydrous ethanol three times, and then lyophilized to obtain two selenylated LP products, namely SeLP1 and SeLP2. In addition, an equal amount of LP was mixed with Na_2_SeO_3_ but without HNO_3_ and treated as above to obtain LP that was used as the control in this study. Afterward, Se contents of the prepared LP, SeLP1, and SeLP2 were detected by an inductively coupled plasma mass spectrometry method [28] using Agilent 7800 ICP-MS inductively coupled plasma mass spectrometry (Agilent Technologies, Santa Clara, CA, USA).

### 2.3. Cell Culture and Assay of Growth Inhibition

As recommended, the cells were cultured in McCoy’s 5A medium supplemented with 10% FBS, 100 U/mL penicillin/streptomycin, and 2.2 g/L NaHCO_3_, and incubated in a humidified incubator with 5% CO_2_ at 37 °C.

The effects of the samples on cell viability were evaluated by a MTT assay, as previously described [18], aiming to determine the suitable doses of the samples [29]. The 100 μL cells were plated into 96-well plates at a density of 1 × 10^5^ cells/well and incubated for 24 h. After discarding the medium, the cells were treated with the medium of 100 μL (negative control), 5-FU of 100 μmol/L (positive control), Na_2_SeO_3_ of 1–16 μg/mL (inorganic Se control), or the three polysaccharide samples (LP, SeLP1, and SeLP2) at 50–800 μg/mL for 24 or 48 h. After medium removal, MTT solution of 100 μL (10 μL 5 mg/mL MTT in 90 μL medium) was added to each well, while the cells were incubated for 4 h. After the medium was discarded, 150 μL of DMSO was added into each well, and a microplate reader (Bio Rad Laboratories, Hercules, CA, USA) was used to determine the optical density of each well at 490 nm. Growth inhibition of the samples on the cells were thus calculated as previously described [18], while the cells from the negative control were regarded with 100% cell viability but without any growth inhibition.

### 2.4. Hoechst 33258 Staining

A fluorescence probe Hoechst 33258 was used for nuclear staining. A total of 2 mL of cells were seeded in 6-well plates (1 × 10^5^ cells/well) and cultured for 24 h. After medium removal, the cells were treated with the medium, or the polysaccharide samples (LP, SeLP1, and SeLP2) at the dose levels of 400–800 μg/mL for 24 and 48 h. After discarding the medium, the cells were washed with PBS twice, fixed with 4% methanol of 0.5 mL at 4 °C for 10 min, washed with PBS thrice again, added to stained 1 mL Hoechst 33258 at 25 °C, and then observed under a fluorescence microscope (ThermoFisher Scientific, Invitrogen EVOS FL Auto 2, Carlsbad, CA, USA) using an objective of 20-fold.

### 2.5. Assay of Cell Colony Formation

A total of 2 mL cells were seeded in 6-well plates (1 × 10^3^ cells/well) and cultured in the medium with 5% FBS for 24 h. Afterward, the cells were treated with the medium or the polysaccharide samples (LP, SeLP1, and SeLP2) at the dose levels of 400–800 μg/mL for 48 h. After discarding the medium, the medium was changed every 3 d until the cells were cultured for 14 and 21 d. The cells were fixed with 1 mL of 4% paraformaldehyde at 4 °C for 15 min, washed with PBS thrice, stained with 1 mL of crystal violet dye at 25 °C for 10 min, and then photographed after drying.

### 2.6. Analysis of Mitochondrial Membrane Potential, Intracellular ROS, and Ca^2+^

A total of 2 mL of cells were seeded into 6-well plates (1 × 10^5^ cells/well), cultured for 24 h, and then treated by the medium or the polysaccharide samples (LP, SeLP1, and SeLP2) at the dose levels of 400–800 μg/mL for 24 and 48 h. To assay the mitochondrial membrane potential (MMP), the treated cells were washed with PBS twice, added to 1 mL JC-1 dye staining solution, and incubated at 37 °C for 20 min. The cells were then seeded into 96-well plates, while fluorescence intensities were determined using the microplate reader. MMP of the treated cells was expressed as the red/green fluorescent ratio as previously described [30].

To assay the intracellular ROS, the treated cells were washed with PBS twice, added to 1 mL DCFH-DA (2′,7′-dichlorodihydrofluorescein diacetate, 5 μmol/L) staining solution, and then incubated at 37 °C for 20 min. The cells were seeded into 96-well plates to detect fluorescence intensities using the microplate reader and excitation/emission wavelengths of 488/525 nm. Relative intracellular ROS levels of the treated cells were expressed as the percentages of the control cells. To assay the intracellular Ca^2+^, the treated cells were washed with Krebs–Ringer buffer (pH 7.4) twice, added to 1 mL of Fura-2 AM (5 μmol/L) staining solution, and incubated at 37 °C for 40 min. The cells were seeded into 96-well plates to measure fluorescence intensities using the microplate reader. Intracellular Ca^2+^ (i.e., [Ca^2+^]_i_) was thus calculated as described previously [31].

### 2.7. Assay of Cell Apoptosis by Flow Cytometry

A total of 2 mL of cells were seeded into 6-well plates (1 × 10^5^ cells/well) and cultured for 24 h, and then treated by the medium or the polysaccharide samples (LP, SeLP1, and SeLP2) at 400–800 μg/mL doses for 24 and 48 h. The cells were washed with PBS twice, added to 195 μL of Annexin V-FITC binding buffer, 5 μL of Annexin V-FITC, and 10 μL of PI staining solution at 25 °C for 10 min, and detected by a flow cytometer (Type BDFACS Aria II, BD Bioscience, Franklin Lakes, NJ, USA) to obtain the respective percentages of the viable, necrotic, early apoptotic, and late apoptotic cells (Q1–Q4).

### 2.8. Reverse Transcription Quantitative Real-Time PCR Assay

A total of 2mL cells were seeded into 6-well plates (1 × 10^5^ cells/well) and cultured for 24 h, and treated by the medium or the polysaccharide samples (LP, SeLP1, and SeLP2) at the dose levels of 400−800 μg/mL for 48 h. The total RNA was extracted according to the procedures of the RNAprep Pure Cell Kit, followed by a reverse transcription of the RNA into complementary DNA (cDNA) using the NovoScript^®^ two-step RT-PCR Kit as well as a final amplification of the cDNA using the NovoScript^®^ SYBR qPCR SuperMix Plus and Biosystems StepOnePlus real-time PCR system (Life Technologies Corp., Carlsbad, CA, USA). PCR data were analyzed and calculated by the classic 2^−ΔΔCt^ method [32]. The primers used in this assay had the sequences listed in Table 1, while the β-actin housekeeping gene was used as the internal standard.

### 2.9. Assay of Western-Blotting

A total of 5 mL cells were cultured in a 25 cm^2^ cell culture flask (1 × 10^6^ cells/flask) for 24 h, and treated by the medium, LP (800 μg/mL), SeLP1 (400 μg/mL), SeLP2 (400 μg/mL), Ac-DEVD-CHO (50 μmol/L), and SeLP2 (400 μg/mL) plus Ac-DEVD-CHO (50 μmol/L) for 48 h. The cells were lysed for 30 min at 4 °C using the radio-immunoprecipitation assay (RIPA) lysis buffer of 100 μL containing PMSF (1 mmol/L), and centrifuged at 12,000× *g* for 5 min, while the total protein concentration was measured by the BCA Protein Analysis Kit. The protein samples were separated on a 12% SDS-PAGE gel and electro-transferred to the nitrocellulose membranes, blocked with 5% skimmed milk for 2 h at 37 °C, incubated with primary antibodies (1:1000 dilution) at 4 °C for 12 h, and incubated with secondary antibodies (1:5000 dilution) at 37 °C for 1 h. An Image Quant LAS 500 (Fujifilm, Tokyo, Japan) was used to detect the protein bands, which were then quantified using ImageJ software version 2x (National Institutes of Health, Bethesda, MD, USA). Moreover, β-actin was used as an endogenous standard to normalize band density.

### 2.10. Statistical Analysis

All reported data were collected from three independent experiments or analyses, and reported as the means ± standard deviations. Significant differences (*p* < 0.05) between the mean values of the individual group were analyzed by the Statistical Program for Social Sciences 16.0 software package (SPSS Inc., Chicago, IL, USA) using Duncan’s multiple range tests.

## 3. Results

### 3.1. Growth Inhibition of Polysaccharide Samples on the Cells

Using the inductively coupled plasma mass spectrometry method, SeLP1 and SeLP2 were detected to have respective Se contents of 1.46 and 4.79 g/kg, while the unreacted LP had much lower Se content of 0.26 μg/kg. The growth suppression of HT-29 cells in response to the polysaccharide samples (LP, SeLP1 and SeLP2), 5-FU, and Na_2_SeO_3_ are shown in Figure 1. The 5-FU at 100 μmol/L could suppress cell growth, resulting in inhibition values of 35.9% (24 h) and 52.2% (48 h). Na_2_SeO_3_ was very toxic to the cells. Na_2_SeO_3_ at higher dose levels, together with longer treatment times (e.g., 16 μg/mL and 48 h), even caused total cell death (i.e., viability value near zero) (Figure 1A). With a cell treatment of 24 h, the inhibition values of LP, SeLP1, and SeLP2 at 50–800 μg/mL were 6.6–12.5%, 10.8–30.3%, and 14.5–39.6%, respectively (Figure 1B). When the cells were treated for 48 h with LP, SeLP1, and SeLP2 at these dose levels, the detected inhibition values were enhanced to 7.4–14.5%, 11.2–45.2%, and 15.5–60.1%, respectively (Figure 1C). Generally, LP, SeLP1, and SeLP2 dose- and time-dependently had anti-proliferative activities to the cells. Additionally, it was not possible to calculate the IC_50_ value of LP with the obtained data, while SeLP1 and SeLP2 could be estimated with respective IC_50_ values of 4440 and 1760 (24 h) or 1240 and 450 μg/mL (48 h), respectively. SeLP1 and especially SeLP2 had higher growth inhibition than LP on the cells, suggesting the used chemical selenylation and higher selenylation extent could enhance the anti-proliferation of the selenylated LP products on the cells.

Hoechst 33258 staining results (Figure 2) confirmed the anti-cancer activities of the polysaccharide samples to the cells. Overall, the treated cells had morphological changes such as the nuclei shrinkage and fragmentation, chromatin condensation, and formation of apoptotic bodies, while SeLP1, and especially SeLP2, were more effective than LP to alter cell morphology. The results of cell colony formation (Figure 3) also showed that SeLP1 and especially SeLP2 had higher proliferation inhibition on the cells than LP. These results consistently proved that the chemical selenylation of LP caused higher activity for SeLP1 and SeLP2, while higher selenylation extent yielded activity enhancement.

### 3.2. Intracellular ROS, Ca^2+^, and MMP Loss in Response to Polysaccharide Samples

When the polysaccharide samples (LP, SeLP1, and SeLP2) were used to treat the cells for 24 and 48 h, ROS levels of the treated cells showed obvious increases (Figure 4A). Compared with the control cells, the LP-treated cells (800 μg/mL dose) showed ROS levels of 108.6% (24 h) and 112.9% (48 h), while the SeLP1-treated cells (400−800 μg/mL doses) caused ROS levels of 119.6–127.1% (24 h) or 125.5–169.4% (48 h). When the cells were exposed to SeLP2 of the two doses, the resultant ROS levels were 129.1–145.4% (24 h) and 175.1–214.1% (48 h).

Compared with the control cells, MMP of the cells treated by the polysaccharide samples were decreased, reflected by the reduced ratios of red/green fluorescence (Figure 4B). In the control cells, the detected ratios were 15.7 (24 h) and 14.3 (48 h). Meanwhile, the LP-treated cells (800 μg/mL dose) showed lower ratios of 11.5 (24 h) and 11.1 (48 h), demonstrating a clear MMP loss. Moreover, SeLP1 and SeLP2 at 400−800 μg/mL caused the respective ratios of 7.3–9.4 and 6.4–7.5 (24 h) or 5.9–8.3 and 2.5–5.8 (48 h), indicating greater MMP loss. Intracellular Ca^2+^ levels of the treated cells also showed obvious changes (Figure 4C). The LP-treated cells showed [Ca^2+^]_i_ values of 103.9% (24 h) and 105.5% (48 h), while the SeLP1-treated cells had [Ca^2+^]_i_ values of 106.5–108.8% (24 h) and 108.1–123.9% (48 h). If the cells were treated by SeLP2, the resultant [Ca^2+^]_i_ values were 109.2–117.2% (24 h) and 125.3–150.3% (48 h). Overall, these results revealed that the used chemical selenylation and higher selenylation extent consistently endowed the selenylated products with higher potentials in the cells, causing enhanced ROS and [Ca^2+^]_i_ levels together with MMP damage. Considering that the unusual levels of ROS and [Ca^2+^]_i_ as well as MMP loss in the cells might lead to cell apoptosis, it was reasonable that the assessed samples might induce cell apoptosis via the mitochondria and ER organelles, which should be clarified.

### 3.3. Apoptosis Induction of Polysaccharide Samples to the Cells

The cells were thus exposed to LP, SeLP1, and SeLP2 for 24 and 48 h, while total apoptotic proportions (i.e., Q2 + Q4, early plus late apoptotic cells) of the treated cells were measured and calculated (Figure 5 and Table 2). In detail, the control cells showed a total apoptotic proportion of 5.1% (24 h) or 5.4% (48 h), while the LP-treated cells (800 μg/mL dose) gave a total apoptotic proportion of 7.9% (24 h) or 8.3% (48 h). Meanwhile, SeLP1 and SeLP2 at 400–800 μg/mL doses caused the respective total apoptotic proportions of 9.5–10.0% and 12.1–16.2% (24 h) or 20.3–31.6% and 33.5–46.2% (48 h). Thus, the selenylated LP products had higher apoptosis induction than the unmodified LP in the cells, while higher selenylation extent also contributed to enhanced apoptosis induction for the selenylated products.

### 3.4. The Expression Changes of the Apoptosis-Related Genes and Proteins in the Cells

To further reveal the anti-cancer activities of LP, SeLP1, and SeLP2 in the cells, mRNA expression of the eight apoptosis-related genes was assayed (Table 3). All assessed samples were able to upregulate or downregulate the eight genes. When the cells were treated by LP, SeLP1, and SeLP2, the respective expression levels of Bax, caspases-3/-8/-9, CHOP, cytochrome c, DR5, and Bcl-2 were 1.0–1.7, 1.0–1.3, 1.0–1.4, 1.0–1.5, 1.0–1.6, 1.0–1.4, 1.0–1.6, and 0.8–1.0-fold, respectively. Overall, the samples upregulated the expression of the seven pro-apoptotic genes Bax, caspases-3/-8/-9, CHOP, cytochrome c, and DR5, but downregulated the mRNA expression of one anti-apoptotic gene Bcl-2. In addition, SeLP1, and especially SeLP2, had higher ability to regulate these genes than LP, suggesting that the chemical selenylation as well as higher selenylation extent enhanced the activities of the selenylated LP products.

The western blotting results indicated that the assessed samples were also capable of upregulating or downregulating the expression of the eight apoptosis-related proteins in the treated cells (Figure 6A and Table 4). In brief, SeLP1 and SeLP2 were more active than LP to upregulate the expression of seven pro-apoptotic proteins Bax, cytochrome c, DR5, CHOP, and cleaved caspases-3/-8/-9 in the cells or to downregulate the expression of anti-apoptotic protein Bcl-2. Additionally, caspase-3 is a key apoptosis enforcer in the cells. SeLP2 showed similar capacity in the cells as 5-FU to downregulate the expression of procaspase-3 but to upregulate the expression of cleaved caspase-3. Furthermore, the caspase-3 inhibitor Ac-DEVD-CHO, well-known for its anti-apoptosis function in cells, did not show any effect on procaspase-3 expression, but antagonized the SeLP2-caused down-expression of procaspase-3 (Figure 6B and Table 4), which proved the apoptosis induction of SeLP2. It was thus inferred that SeLP1 and SeLP2 could exert apoptosis induction to HT-29 cells via an activation of the death receptor, mitochondrial-dependent, and ER stress pathways (Figure 7), because they possessed a capacity to upregulate the related proteins such as cytochrome c, DR5, CHOP, and cleaved caspases-3/-8/-9 in the cells simultaneously.

## 4. Discussion

Accumulating evidence indicates that increased intake of Se from various food resources could reduce the incidence of cancers, while Se-containing polysaccharides or proteins as two typical organic Se-containing substances have been widely assessed for their potentials to prevent cancers [6,7,8]. Selenylated polysaccharides are regarded as promising chemopreventive agents for tumor because they can modulate cell growth, apoptosis, and metastasis by targeting a variety of molecules and biochemical pathways related to tumor development [33,34,35]. Past results have shown that the selenylated polysaccharides from *G. frondosa* could cause strong anti-cancer effect on the tumor-bearing mice by improving the vital immune function [18], while the Se-containing polysaccharides from the root of *Astragalus membranaceus* or *Ginkgo biloba* were capable of inhibiting the proliferation of H22 ascite liver cancer and S180 sarcoma cells [36], or by inducing cell apoptosis in human bladder cancer T24 cells via the mitochondrial-dependent pathway [37]. In addition, the chemical selenylation was also proven to enhance the anti-tumor activity of peptidoglycans because the selenylated peptidoglycans were observed with stronger anti-proliferation on HT-29 cells [38]. It was also verified that the selenylated *A. sphaerocephala* polysaccharides with Se contents of about 4344 and 13,030 mg/kg had anti-tumor potentials, which were positively correlated with the Se contents [29]. These published results thus proved that SeLP1 and SeLP2 had higher activities to the cells than LP, and the performed chemical selenylation, together with the resultant selenylation extent, made a contribution to the assessed anti-cancer activities of the two selenylated products. Thus, this investigated chemical modification has potential application to alter polysaccharide bioactivity.

Cell apoptosis is one of the important mechanisms for inhibiting cancer cell growth. Cell apoptosis is triggered by three pathways, namely death receptor, mitochondrial apoptotic, and ER stress pathways. In general, organic Se compounds such as selenylated polysaccharides are regarded to induce the apoptosis of cancer cells effectively [39,40,41]. For example, the selenylated polysaccharides from *Artemisia sphaerta* induced HepG-2 cell apoptosis by activating the death receptor pathway via upregulating apoptotic protein caspase-8 [29], while the Se-enriched *Ganoderma lucidum* could induce the apoptosis of human breast cancer MCF-7 cells via MMP disruption and activating the mitochondrial apoptosis pathway with upregulated apoptotic protein caspases-3/-8/-9 [5]. Consistent with these mentioned studies, this study also revealed that the assessed polysaccharide samples could exert anti-cancer effects, reflected by the growth suppression, enhanced ROS formation, MMP loss, and more importantly, the apoptosis induction via the mitochondrial-dependent and death receptor pathways. However, few researchers have paid attention to reveal whether the ER stress pathway is also one of the potential mechanisms of the selenylated polysaccharides to induce cell apoptosis. A previous study reported that methylseleninic acid had apoptosis induction to human prostate cancer cells (PC-3) via ER-mediated stress [42]. Thus, the role of selenylated polysaccharides in ER-mediated apoptosis induction requires further investigation. It has been regarded that a higher level of ROS in cells induce ER stress; subsequently, ER stress leads to the release of large amounts of Ca^2+^ into the cytoplasm, while the overload of Ca^2+^ then causes MMP loss [31]. It was reasonable in this study that the assessed samples could induce the ER-mediated cell apoptosis because they obviously showed capacity to enhance Ca^2+^ level and upregulated the related proteins like CHOP in the treated cells.

Chemical modifications can endow the polysaccharides with greater anti-cancer properties. According to the literature, the sulfated polysaccharides from ginseng have anti-tumor activity to HCT-116, HepG-2, and MCF-7 cells [43], while the acetylated galactan from *Ophiopogon japonicus* could induce the apoptosis of pancreatic cancer (BxPC-3) cells [44]. Moreover, the carboxymethylated polysaccharides from *Poria cocos* showed growth inhibition on SGC-7901 and HT-29 cells [45], while the phosphorylated levan from *Bacillus licheniformis* was able to inhibit tumor cell proliferation and induce cell cycle arrest [46]. From a chemical point of view, chemical modification can change the types of chemical groups in the polysaccharides and therefore cause altered polysaccharide activities to the cancer cells [47]. The performed chemical selenylation of LP induced the −SeO_3_H groups into LP molecules, while selenite was detected with higher activity to HT-29 cells (Figure 1). It is thus reasonable that the chemical selenylation of natural polysaccharides such as LP resulted in activity increases in HT-29 cells. Meanwhile, the activities of other compounds may be altered due to the performed chemical modifications; for example, polyphenols with covalent modifications such as hydroxylation, methylation, and acylation were detected to have higher anti-cancer activities to HCT-116 and HT-29 cells [48,49,50]. It was also reported that 3′-hydroxypterostilbene was more potent than pterostilbene to inhibit cell growth and induce cell apoptosis to three human colon cancer (e.g., COLO 205, HCT-116, and HT-29) cells [49], while dimethoxycurcumin was more effective than curcumin to inhibit the growth of HCT-116 cells and induce apoptosis [50]. Thus, chemical modifications might endow natural substances with greater activities in the cells. More importantly, two past studies have indicated that selenylated polysaccharides had better anti-cancer effects than Se or polysaccharides [51,52]. It was thus concluded that both SeLP1 and SeLP2 received covalent Se conjugation in their molecules, and finally possessed higher anti-cancer effects on the targeted HT-29 cells than the unmodified LP. However, future investigation into the in vivo anti-colon cancer potential of the selenylated LP products and related molecular mechanism is necessary. Meanwhile, the structural features of LP and SeLP also need to be clarified, which might be used to reveal the possible structure–activity relationship of SeLP.

## 5. Conclusions

The results of this study demonstrated that the chemical selenylation of LP using the Na_2_SeO_3_–HNO_3_ system could conjugate a minor nutrient Se into LP molecules efficiently. Due to the covalent conjugation of −SeO_3_H groups into LP molecules, the obtained selenylated products possessed higher anti-cancer effects on the HT-29 cells. Overall, the selenylated products were more effective than LP in suppressing cell growth, altering cell morphology, enhancing intracellular ROS and Ca^2+^ levels, and causing MMP loss. Furthermore, the selenylated products were more active than LP to induce cell apoptosis by regulating the eight apoptosis-related genes and proteins as well as mediating the death receptor, mitochondrial-dependent, and ER stress pathways simultaneously. Additionally, higher selenylation extent endowed the selenylated product with higher anti-cancer effects on the cells. It is also encouraged to develop the Se-rich food ingredients through chemical approaches including selenylation to enhance their health care functions in the body, and more importantly, the specific chemical features of the modified products should be clarified to reveal their possible structure–activity relationship.

## Figures and Tables

**Figure 1 nutrients-14-01710-f001:**
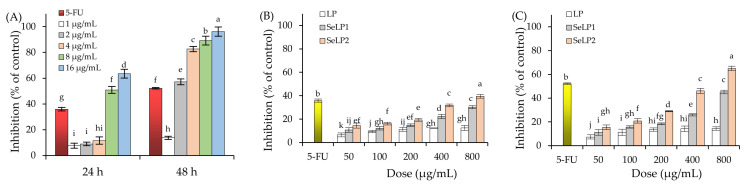
Growth inhibition of 5-Fu (100 μmol/L) and Na_2_SeO_3_ (**A**) as well as longan polysaccharides (LP) and two selenylated polysaccharides (SeLP1 and SeLP2) on HT-29 cells with exposure times of 24 (**B**) and 48 h (**C**). Different lowercase letters above the columns indicate significant differences (*p* < 0.05).

**Figure 2 nutrients-14-01710-f002:**
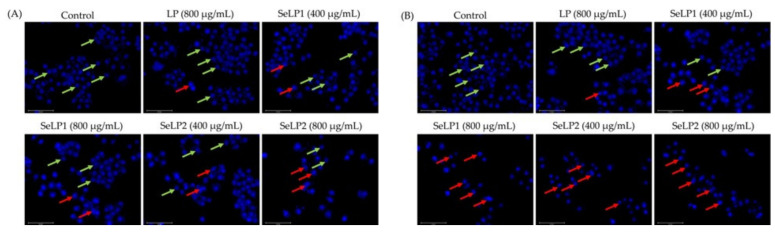
Morphological features of HT-29 cells treated with cell medium (control), longan polysaccharides (LP), and selenylated polysaccharides (SeLP1 and SeLP2) for 24 h (**A**) and 48 h (**B**). The green arrows indicate normal cells, while the red ones indicate the cells with morphological alteration.

**Figure 3 nutrients-14-01710-f003:**
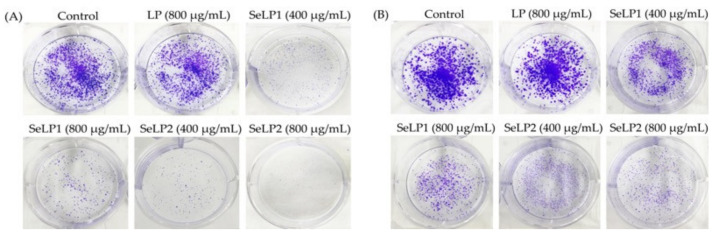
Long-term anti-proliferation of longan polysaccharides (LP) and the two selenylated polysaccharides (SeLP1 and SeLP2) on HT-29 cells with treatment times of 14 (**A**) and 21 days (**B**).

**Figure 4 nutrients-14-01710-f004:**
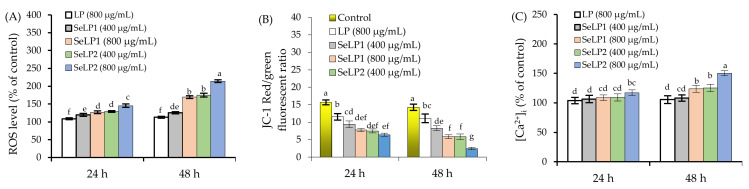
Effects of longan polysaccharides (LP) and selenylated polysaccharides (SeLP1 and SeLP2) on intracellular ROS (**A**), mitochondrial membrane potential (MMP) loss (**B**), and intracellular Ca^2+^ (**C**) of HT-29 cells. Different lowercase letters above the columns indicate significant differences (*p* < 0.05).

**Figure 5 nutrients-14-01710-f005:**
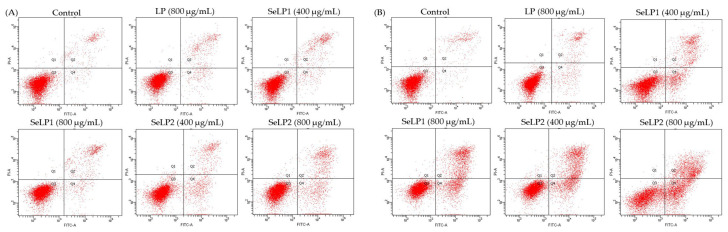
Apoptosis induction of longan polysaccharides (LP) and the selenylated polysaccharides (SeLP1 and SeLP2) toward HT-29 cells with treatment times of 24 (**A**) and 48 h (**B**). Cells in Q1–Q4 represent the necrotic, late apoptotic, intact, and early apoptotic cells, respectively.

**Figure 6 nutrients-14-01710-f006:**
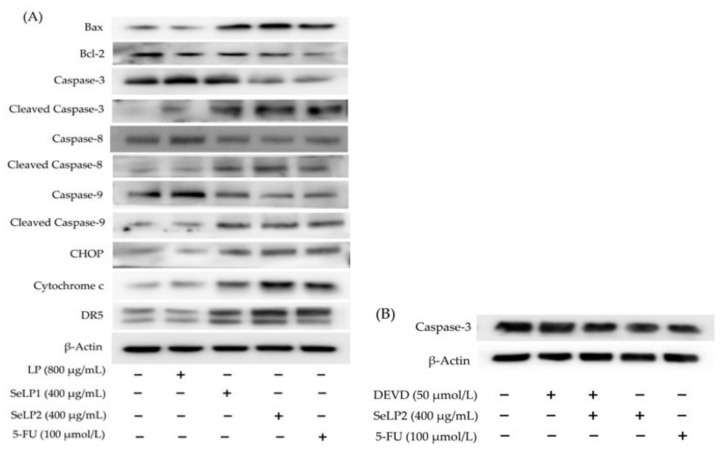
Western blot assay of the apoptosis-related proteins in HT-29 cells treated without or with 5-FU, longan polysaccharides (LP), and two selenylated polysaccharides (SeLP1 and SeLP2) for 48 h (**A**), or the caspase-3 in HT-29 cells treated with a caspase-3 inhibitor DEVD for 48 h (**B**). “+” indicates HT-29 cells were treated with this sample, “−“ indicates HT-29 cells were not treated with this sample.

**Figure 7 nutrients-14-01710-f007:**
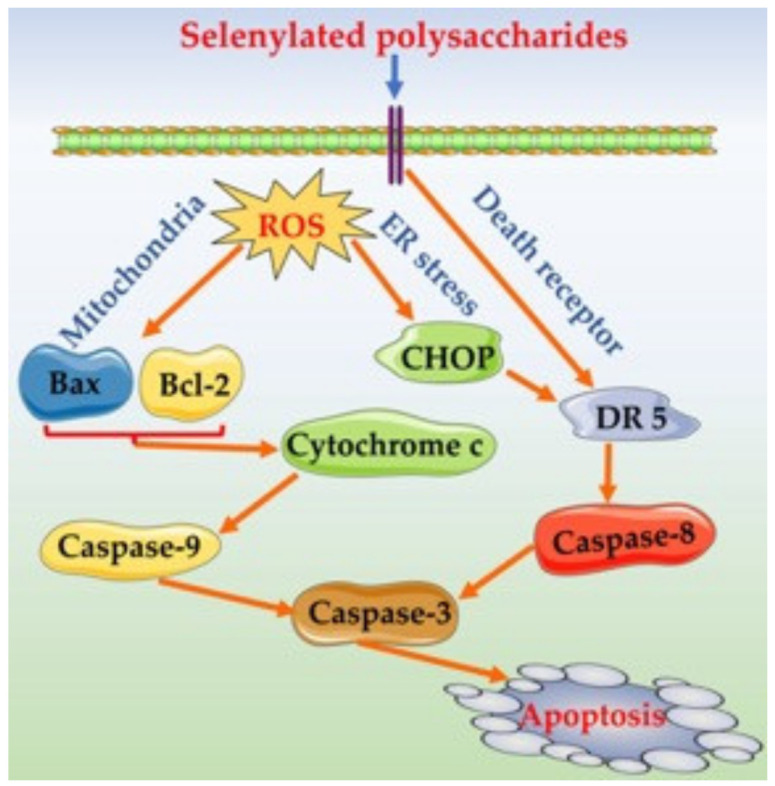
The underlying mechanism for the selenylated polysaccharides to induce cell apoptosis.

**Table 1 nutrients-14-01710-t001:** Primer sequences used in real-time PCR.

Genes	Primer Sequences (5′-3′)	Species
Bax	Forward: 5′-CCAGAGGCGGGGGATGATT-3′ Reverse: 5′-CTGATCAGTTCCGGCACCTT-3′	Human
Bcl-2	Forward: 5′-CTTTGAGTTCGGTGGGGTCA-3′ Reverse: 5′-GGGCCGTACAGTTCCACAAA-3′	Human
Caspase-3	Forward: 5′-TTGAGACAGACAGTGGTGTTGATGATG-3′ Reverse: 5′-ATAATAACCAGGTGCTGTGGAGTATGC-3′	Human
Caspase-8	Forward: 5′-CAAACTTCACAGCATTAGGGAC-3′ Reverse: 5′-ATGTTACTGTGGTCCATGAGTT-3′	Human
Caspase-9	Forward: 5′-CTGCTGCGTGGTGGTCATTCTC-3′ Reverse: 5′-CACAATCTTCTCGACCGACACAGG-3′	Human
CHOP	Forward: 5′-TAAAGATGAGCGGGTGGCAG-3′ Reverse: 5′-CTGCCATCTCTGCAGTTGGA-3′	Human
Cytochrome c	Forward: 5′-GAGTAATAATTGGCCACTGCCT-3′ Reverse: 5′-AATCAGGACTGCCCAACAAAA-3′	Human
DR5	Forward: 5′-CTGATCACCCAACAAGACCTAG-3′ Reverse: 5′-GATGCAATCTCTACCGTCTTCT-3′	Human
β-Actin	Forward: 5′-CCACCATGTACCCTGGCAT-3′ Reverse: 5′-ACTCCTGCTTGCTGATCCAC-3′	Human

**Table 2 nutrients-14-01710-t002:** The measured total apoptotic cells (%, Q2 + Q4) for HT-29 cells exposed to longan polysaccharides (LP) and two selenylated polysaccharides (SeLP1 and SeLP2).

Cell Group	Total Apoptotic Proportion (%)	
	24 h	48 h
Control	5.1 ± 0.1 ^e^	5.4 ± 0.3 ^d^
LP (800 μg/mL)	7.9 ± 0.7 ^d^	8.3 ± 0.2 ^d^
SeLP1 (400 μg/mL)	9.5 ± 0.4 ^c^	20.2 ± 0.7 ^c^
SeLP1 (800 μg/mL)	10.0 ± 0.9 ^c^	31.6 ± 1.9 ^b^
SeLP2 (400 μg/mL)	12.1 ± 0.4 ^b^	33.5 ± 2.8 ^b^
SeLP2 (800 μg/mL)	16.2 ± 1.7 ^a^	46.2 ± 3.2 ^a^

Different lowercase letters after the data as the superscripts indicate significant differences (*p* < 0.05).

**Table 3 nutrients-14-01710-t003:** Expression changes of the eight apoptosis-related genes in HT-29 cells exposed to longan polysaccharides (LP) and two selenylated polysaccharides (SeLP1 and SeLP2).

Gene	Cell Group and Relative Expression Fold					
	Control	LP (800 μg/mL)	SeLP1 (400 μg/mL)	SeLP1 (800 μg/mL)	SeLP2 (400 μg/mL)	SeLP2 (800 μg/mL)
Bax	1.0 ± 0.1	1.0 ± 0.1	1.3 ± 0.1	1.4 ± 0.1	1.5 ± 0.1	1.7 ± 0.1
Bcl-2	1.0 ± 0.1	1.0 ± 0.0	0.9 ± 0.1	0.8 ± 0.1	0.8 ± 0.1	0.8 ± 0.0
Caspase-3	1.0 ± 0.2	1.0 ± 0.1	1.1 ± 0.0	1.2 ± 0.0	1.2 ± 0.0	1.3 ± 0.1
Caspase-8	1.0 ± 0.1	1.1 ± 0.1	1.2 ± 0.0	1.3 ± 0.1	1.3 ± 0.1	1.4 ± 0.1
Caspase-9	1.0 ± 0.1	1.0 ± 0.1	1.2 ± 0.1	1.3 ± 0.0	1.4 ± 0.0	1.5 ± 0.0
CHOP	1.0 ± 0.0	1.1 ± 0.1	1.1 ± 0.0	1.4 ± 0.0	1.4 ± 0.1	1.6 ± 0.1
Cytochrome c	1.0 ± 0.1	1.1 ± 0.0	1.2 ± 0.1	1.3 ± 0.1	1.3 ± 0.1	1.4 ± 0.1
DR5	1.0 ± 0.1	1.0 ± 0.1	1.2 ± 0.0	1.5 ± 0.0	1.5 ± 0.1	1.6 ± 0.1

**Table 4 nutrients-14-01710-t004:** Expression changes of the apoptosis-related proteins in HT-29 cells exposed to longan polysaccharides (LP), two selenylated polysaccharides (SeLP1 and SeLP2), inhibitor DEVA, or DEVA plus SeLP2.

Protein	Relative Expression Fold						
	Control	LP	SeLP1	SeLP2	5-FU	DEVD	DEVD and SeLP2
Bax	1.0 ± 0.2	1.0 ± 0.2	1.9 ± 0.2	2.3 ± 0.2	2.0 ± 0.1	NA	NA
Bcl-2	1.0 ± 0.1	1.0 ± 0.1	0.7 ± 0.1	0.6 ± 0.1	0.6 ± 0.1	NA	NA
CHOP	1.0 ± 0.1	1.1 ± 0.3	2.0 ± 0.3	2.1 ± 0.2	1.9 ± 0.1	NA	NA
Cleaved Csapase-3	1.0 ± 0.0	1.4 ± 0.1	1.6 ± 0.2	2.2 ± 0.2	1.8 ± 0.2	NA	NA
Cleaved Caspase-8	1.0 ± 0.0	1.0 ± 0.3	2.0 ± 0.1	2.1 ± 0.2	1.9 ± 0.2	NA	NA
Cleaved Caspase-9	1.0 ± 0.1	1.1 ± 0.3	2.0 ± 0.3	2.4 ± 0.3	2.4 ± 0.1	NA	NA
Cytochrome c	1.0 ± 0.2	1.3 ± 0.2	1.9 ± 0.1	2.4 ± 0.2	1.8 ± 0.1	NA	NA
DR5	1.0 ± 0.0	1.1 ± 0.1	1.9 ± 0.1	2.0 ± 0.1	1.9 ± 0.2	NA	NA
Procaspase-3	1.0 ± 0.2	NA	NA	0.6 ± 0.1	0.6 ± 0.0	1.1 ± 0.2	0.8 ± 0.0

NA, not assessed.

## Data Availability

All data are contained within the article.

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
