# Peer review of "Enhanced Growth Inhibition and Apoptosis Induction in Human Colon Carcinoma HT-29 Cells of Soluble Longan Polysaccharides with a Covalent Chemical Selenylation"

_nutrients, 2022, doi:10.3390/nu14091710_

Round 1

Reviewer 1 Report

In this paper by Yu et al., the Authors investigated the apoptotic effect of a chemical selenylation of soluble polysaccharides from logan fruit in HT29 cell. The results have shown the the cytotoxic and apoptotic effects were depending by time e concentration and that selenylation was effective to increase the biological effect of LP.

The study is well conducted, the methodologies are pertinent, the manuscript is well written, and the overall results may improve the knowledge of the biological effect of selenium.

Despite this consideration some issue raised.

Major revision.

The Authors stated that Se content in SeLP1 and SeLP2 ere 1.46g/kg and 4.79g/kg, respectively (line 232) without giving any explanation how Se was measured or if those values were a theorical estimation. Please explain.

The Authors should indicate if the selenium concentration supplemented to intestinal cell is in a range of physiologic and it is comparable to Se concentration could be obtained after a meal or use of supplements

The antioxidant activity of Se is well known but in this study the Authors revealed an increase of ROS production after SeLP in a time and concentration dependent manner (Figure 4A). Please justify

The Authors should explain why they have not included the supplementation with Na2SeO3 also to the apoptotic experiments. This could discriminate the effect of Se bounded to LP from the effect “per sè”. This was particularly evident in a previous paper (doi:10.3390/nu10121898).

The discussion session includes mainly a list of previous studies without giving substantial information about the mechanism at the basis of the results obtained. Please rewrite it.

Conclusion: please include a possible exploitation of the results

Minor revision:

Title: the term “bioactivity” in this contest is too generic.

Figure 1A. Please include legend also for 16ug/ml concentration. Figure 1B and 1C. Please include legend also for 5-FU

Author Response

In this paper by Yu et al., the Authors investigated the apoptotic effect of a chemical selenylation of soluble polysaccharides from logan fruit in HT29 cell. The results have shown the cytotoxic and apoptotic effects were depending by time e concentration and that selenylation was effective to increase the biological effect of LP.

Reply: Thanks for your kindly comment.

The authors had studied the bioactivities of natural polysaccharides with chemical selenylation in the past years, and found that this selenylation caused higher immune regulation and anti-cancer effects in model cells or animals. The authors are pleased that the reviewers gave their kindly and valuable suggestions for our past works.

The study is well conducted, the methodologies are pertinent, the manuscript is well written, and the overall results may improve the knowledge of the biological effect of selenium.

Reply: Thanks for your kindly comment.

The authors carefully designed the assessed experiments by using these reported methods or conditions, and have made their best effort to prepare this manuscript. Thanks again!

Despite this consideration some issue raised.

Major revision.

1, The Authors stated that Se content in SeLP1 and SeLP2 ere 1.46g/kg and 4.79g/kg, respectively (line 232) without giving any explanation how Se was measured or if those values were a theorical estimation. Please explain.

Reply: A good suggestion.

The Se contents of LP, SeLP1 and SeLP2 were detected directly by the inductively coupled plasma mass spectrometry method, using Agilent 7800 ICP-MS inductively coupled plasma mass spectrometry.

Based on your suggestion, we added this information in the M&M section. Please see the added sentences in the lines 148-151 of the revised manuscript.

Thanks!

2, The Authors should indicate if the selenium concentration supplemented to intestinal cell is in a range of physiologic and it is comparable to Se concentration could be obtained after a meal or use of supplements

Reply: A good comment.

In this study, the human colon carcinoma HT-29 cells were used as cell model. Based on a reported study from other researchers (please see the published paper below), we selected sample doses of 50-800 mg/mL in our experiments to determine growth inhibition of the samples on the cells.

Liu, S.; Hu, J.H.; Li, M.; Zhu, S.Y.; Guo, S.J.; Guo, H.Y.; Wang, T.; Zhang, Y.D.; Zhang, J.; Wang, J.L. The role of Se content in improving anti-tumor activities and its potential mechanism for selenized Artemisia sphaerocephala polysaccharides. Food Funct. 2021, 12, 2058–2074.

In the present, we can not ensure that these sample doses are in the physiologic levels. Two considerations of us are listed here. First, we still do not measure how many inorganic Se might be released during the digestion of these samples in the digestive tract. Second, as you know, the daily intake of dietary fibers in people varies obviously. Thus, a direct calculation might lead to incorrect conclusion.

However, we can make an indirectly estimation for the present results. As you know, the LP are non-digestible in the body, thus would enter the colon without absorption. It is thus assumed that the selenylated LP will exist in the feces. In general, the average feceal volume of normal Asian adults is about 0.3 kg; thus, 50-800 mg/mL sample doses will nearly equal to 15-240 mg sample amount. Because the selenylated samples contain Se of 1.46 and 4.79 g/kg (or Se of 1.46-4.79 mg/mg), the estimated Se amount therefore range from 1.46*15=21 (the lowest) to 4.79*240=1150 mg (the highest).

The recommended daily intake (RDI) of Se is about 50-250 mg/d for adults. It is thus estimated that the lower sample doses (50-200 mg/mL for SeLP2) or lower selenylation extent (SeLP1) thus might fail into this RDI range. However, we do not agree to compare these data, because practical absorption of the selenylated polysaccharides has not measured, which might be expected with very lower bioavailability.

Thanks!

3, The antioxidant activity of Se is well known but in this study the Authors revealed an increase of ROS production after SeLP in a time and concentration dependent manner (Figure 4A). Please justify

Reply: Thanks for your kindly comment.

Yes, this study observed that the treated cells had enhanced ROS production. A previous study also has found that Se-containing compounds could deplete GSH in cancer cells, thereby induce high ROS production (please see the published paper below).

Li, T.Y.; Li, F.; Xiang, W.T.; Yi, Y.; Chen, Y.Y.; Cheng, L.; Liu, Z.; Xu, H.P. Selenium-containing amphiphiles reduced and stabilized gold nanoparticles: kill cancer cells via reactive oxygen species. ACS Applied Materials & Interfaces 2016, 8, 22106-22112.

Moreover, natural oxidants like polyphenols have abilities to promote ROS formation in several cancer cells. Due to too many reference works had been published yet, we do not give detail examples here.

We measured ROS levels in the cells, aiming to reveal the ER-mediated apoptosis. Thanks!

4, The Authors should explain why they have not included the supplementation with Na2SeO3 also to the apoptotic experiments. This could discriminate the effect of Se bounded to LP from the effect “per sè”. This was particularly evident in a previous paper (doi:10.3390/nu10121898).

Reply: Thanks for your kindly comment.

Na2SeO3 was very toxic to the cells. Please see our results given in the lines 254-256 and Figure 1A. The obtained results were used to confirm that the unreacted Na2SeO3 was efficiently removed by the ethanol; otherwise, SeLP1 and SeLP2 should have very higher growth inhibition.

Moreover, our aim was to reveal whether the used selenylation could cause higher activity for SeLP1 and SeLP2. Thus, LP must be used as a control. Based on this consideration, these results given in Figures 2-6 and Tables 2-3 did not contain the results using Na2SeO3.

In addition, the mentioned paper (doi:10.3390/nu10121898) has been cited in the revised manuscript. Please our revisions in the lines 64-66 of the revised manuscript. This published work can give background of our work. Thanks again!

5, The discussion session includes mainly a list of previous studies without giving substantial information about the mechanism at the basis of the results obtained. Please rewrite it.

Reply: Good.

We made a major revision for this section. Please see the lines 404-408, 414-416, and 428-432. ThankI

6, Conclusion: please include a possible exploitation of the results

Reply: Yes.

We made a minor revision here. Please see the lines 448-449 and 456-458 in the revised manuscript. Thank!

Minor revision:

7, Title: the term “bioactivity” in this contest is too generic.

Reply: Thanks for your kindly comment.

We made a minor revision for the original paper title. The revised paper title now is shown as “Enhanced growth inhibition and apoptosis induction in human colon carcinoma HT-29 cells of soluble longan polysaccharides with a covalent chemical selenylation”.

8, Figure 1A. Please include legend also for 16 ug/ml concentration. Figure 1B and 1C. Please include legend also for 5-FU.

Reply: Thanks for your kindly comment.

The result using 16 ug/ml of Na2SeO3 has been included in Figure 1A. As you see, we can not show the result of 5-Fu or Na2SeO3 again in the Figure 1B and 1C. The journal editor surely rejects such revision, because this treatment leads to repeated data report. Thanks!

Reviewer 2 Report

The manuscript concerns a problem described for several times by researchers specializing in the analysis of the influence of selenium incorporation into the structure of polysaccharides on their biological, often anti-cancer activity.

The big advantage of work is that the tests of biological activity conducted by the Authors are accurate and performed at a satisfactory level.

On the other hand, a serious disadvantage of the reviewed work is a low degree of novelty of scientific research and lack of more precise data on the structure of the studied compounds.

-The authors use the method of selenation of polysaccharides by esterification with selenious acid (sodium selenite in an acidic environment), described many times and therefore not innovative.

- to the selenylation are subjected fractions of undefined composition.

No, even quite basic, studies of the structure of the isolated polysaccharides were conducted. Therefore, fractions of undefined composition are subjected to the selenylation. If the described by the Authors soluble polysaccharides isolated from longan fruit have been previously subjected to the structural studies, appropriate references should be provided.

- There are no references to the enzymatic method of isolating the polysaccharide fractions. Has it been used previously, by whom?

Numerous expressions used by the Authors seem to be not correct (example - lines 410-412?).

The conclusion (lines 417 et seq.) is confusing: numerous works have already been written on this subject, so it is not an innovative conclusion.

The manuscript requires significant changes (supplements), both literature and experimental data.

The Discussion and Conclusion sections require extensive additions and reformatting. Correction by a native speaker is advisable. 

Author Response

The manuscript concerns a problem described for several times by researchers specializing in the analysis of the influence of selenium incorporation into the structure of polysaccharides on their biological, often anti-cancer activity.

The big advantage of work is that the tests of biological activity conducted by the Authors are accurate and performed at a satisfactory level.

Reply: Thanks for your kindly comment.

Yes, the effect of Se incorporation into the polysaccharides on their biological (more importantly anti-cancer) activities has been investigated in the past studies; however, if this incorporation also leads to increased activities for LP is still not studied. In addition, whether the obtained selenylation extent has possible impact on anti-cancer effect is critical, but unfortunately is not efficiently assessed.

Thus, the present study investigated how covalent Se conjugation endowed LP with higher anti-cancer effects on HT-29 cells, and whether higher selenylation extent yielded higher activities to the cells. In our personal opinion, thus study also has it scientific merits.

Thanks!

1, On the other hand, a serious disadvantage of the reviewed work is a low degree of novelty of scientific research and lack of more precise data on the structure of the studied compounds.

Reply: A good suggestion.

The monosaccharide and possible saccharide linkages were investigated in our work; however, we prefer reporting these data in another manuscript and mainly focusing our object of this manuscript on whether this selenylation brought about activity changes in the cell model.

Thanks!

-The authors use the method of selenation of polysaccharides by esterification with selenious acid (sodium selenite in an acidic environment), described many times and therefore not innovative.

Reply: Thanks for your kindly comment. Yes, the mentioned mechanism is correct.

This selenylation has been used in other studies. However, this study has its merits like using ethanol to remove the unreacted Na2SeO3/H2SeO3 and clarifying how selenylation extent has an impact on activities of the selenylated LP.

In addition, to the best of our knowledge, less attention is paid on ER stress of selenylated polysaccharides in the cells. This study revealed that the samples could induce the ER-mediated cell apoptosis. Thus, this study might give the readers more information about anti-cancer pathways of selenylated polysaccharides.

Thank again!

2, - to the selenylation are subjected fractions of undefined composition.

Reply: Thanks for your kindly comment.

Yes, this study used whole LP for the selenylation. Pease see our reply for the comment #3, in which we answered why we used whole LP for the selenylation.

Moreover, a detailed chemical determination of the final products as well as saccharide fractions is the aim of forthcoming studies.

3, No, even quite basic, studies of the structure of the isolated polysaccharides were conducted. Therefore, fractions of undefined composition are subjected to the selenylation. If the described by the Authors soluble polysaccharides isolated from longan fruit have been previously subjected to the structural studies, appropriate references should be provided.

Reply: Thanks for your kindly comment.

Other researchers had detected the monosaccharides and their linkages in LP. We have added the detailed information in the lines 84-87 of the revised manuscript, and cited this published paper in the revised manuscript.

Original aim of our work is to assume that Se supplementation of longan tree might result in Se fortification in longan fruits and changed bioactivities in longan polysaccharides. Thus, a separation of the polysaccharides into single saccharide fraction is unsuitable because we can not ensure that only one or two polysaccharide fraction will be selenylated, while other fractions are un-reactive to this selenylation. Finally, we had to using a selenylation of whole longan polysaccharides.

4, - There are no references to the enzymatic method of isolating the polysaccharide fractions. Has it been used previously, by whom?

Reply: A good idea.

 We made a minor revision here, to show the used conditions for LP extraction (by citing one reference) and enzymatic treatment (by citing another reference). Please see the revised sentences in the lines 131-135 of the revised manuscript.

Thanks!

5, Numerous expressions used by the Authors seem to be not correct (example - lines 410-412?).

Reply: Thanks for your kindly comment.

We made our best effort to revise and correct paper writing. As you see, many sentences in the revised manuscript were corrected. Due to too many revision have been done, we do not list detailed revision information here.

Thanks again!

6, The conclusion (lines 417 et seq.) is confusing: numerous works have already been written on this subject, so it is not an innovative conclusion.

Reply: Yes, we agree your comment.

We made a minor revision for the conclusion section. Please see the revisions in the lines 446-458. Thanks!

7, The manuscript requires significant changes (supplements), both literature and experimental data.

Reply: Thanks for your kindly comment.

First, we have cited 5 new references into the revised manuscript. Second, we enforced these descriptions about study background and result discussion. Overall, the revised manuscript has added with about 800 words.

The present data effectively confirm the anti-cancer effects of the samples on the cells. The revised manuscript has 4 tables and 7 figures, together with 14 publication pages. Thus, an addition of other experimental data might be unnecessary to the present manuscript.

8, The Discussion and Conclusion sections require extensive additions and reformatting. Correction by a native speaker is advisable. 

Reply: Thanks for your kindly comment.

This manuscript was revised by each coauthor. Each section (especially the introduction, results, and discussion sections) has been carefully read and corrected. We hope this revision could improve paper quality and make this manuscript more readable to the readers.

Round 2

Reviewer 1 Report

The paper is improved

Author Response

The paper is improved

Reply: The authors thank the reviewer for her/his valuable work paid for this manuscript.

Reviewer 2 Report

I cannot fully agree with the Authors' answers to questions 2 and 3.

I take the claim that the structural research will be the subject of further publications. In my opinion, such a statement should therefore appear in the Discussion or Conclusion sections.

However, it is not true that - according to the authors' claim: Original aim of our work is to assume that Se supplementation of longan tree might result in Se fortification in longan fruits and changed bioactivities in longan polysaccharides. Thus, a separation of the polysaccharides into single saccharide fraction is unsuitable because we can not ensure that only one or two polysaccharide fraction will be selenylated, while other fractions are un-reactive to this selenylation. Finally, we had to using a selenylation of whole longan polysaccharides.

As I understand it, the authors are looking for a preparation with an anti-cancer effect - this is what the Introduction section shows.

Even in the group of medicines of natural origin (e.g. herbal medicines), each medicinal preparation should have a defined active ingredient and its content. It is therefore necessary to define the active ingredient (s) in the mixture.

The preparation described by the authors is not strictly of plant origin - it is chemically modified. It is therefore all the more necessary to define the active ingredients.

I understand that the Authors at the current stage of the research were not able to undertake such complex analytical research (polysaccharide separation, e.g. by GPLC methods, 2DNMR structure analysis, etc.), nevertheless, they should at least declare that such research is planned in the future.

Author Response

I cannot fully agree with the Authors' answers to questions 2 and 3.

I take the claim that the structural research will be the subject of further publications. In my opinion, such a statement should therefore appear in the Discussion or Conclusion sections.

Reply: Thank you for the kindly comments on our study.

Yes, specific structure of these polysaccharide samples (LP and SeLP) should be determined in the future. This is important, and we are doing this work.

The suggested issue might be the topic of another study. As you know, several previous studies have assessed some structural features of LP, such as the functional groups, sugar units, polymerization degrees, glycosidic bonds, and conformation. Thus, we aimed to determine whether SeLP might have structural changes.

Based on your suggestion, we made a revision for Discussion section. Please see the lines 445-447. Meanwhile, we also made a revision in the conclusion section; please see the lines 461-463.

Thanks again for your comments and suggestions!

However, it is not true that - according to the authors' claim: Original aim of our work is to assume that Se supplementation of longan tree might result in Se fortification in longan fruits and changed bioactivities in longan polysaccharides. Thus, a separation of the polysaccharides into single saccharide fraction is unsuitable because we can not ensure that only one or two polysaccharide fraction will be selenylated, while other fractions are un-reactive to this selenylation. Finally, we had to using a selenylation of whole longan polysaccharides.

Reply: Yes, it was our aim.

As you see, the used chemical selenylation is a covalent reaction to the –OH groups of the saccharides, thus does not have significant specificity to the saccharide molecules. Thus, using single fraction is unnecessary to the present study. A study of the selenylation of the specified saccharide molecules might be necessary when the saccharide molecules have been characterized with very higher activity to the cells than other ones.

Thanks!

As I understand it, the authors are looking for a preparation with an anti-cancer effect - this is what the Introduction section shows.

Even in the group of medicines of natural origin (e.g. herbal medicines), each medicinal preparation should have a defined active ingredient and its content. It is therefore necessary to define the active ingredient (s) in the mixture.

Reply: Thanks for your kindly comment.

Yes, you are right. We used these classic treatments to prepare LP or SeLP. Thus, the active component of LP and SeLP were polysaccharides.

The preparation described by the authors is not strictly of plant origin - it is chemically modified. It is therefore all the more necessary to define the active ingredients.

Reply: Thanks for your kindly comment.

The used LP preparation follows these reported reference methods, and surely LP belong to phytochemicals of plant origin.

Selenylation occurs in plants, resulting in the natural products like selenylated proteins and polysaccharides. In this study, the chemical selenylation was used to enhance selenylation extent of LP. SeLP1 and SeLP2 also belong to phytochemicals of plant origin.

Thanks!

I understand that the Authors at the current stage of the research were not able to undertake such complex analytical research (polysaccharide separation, e.g. by GPLC methods, 2DNMR structure analysis, etc.), nevertheless, they should at least declare that such research is planned in the future.

Reply: A good idea!

Yes, chemical features of these samples are critical to our whole study. We can not ignore this importance. Due to the present manuscript had sufficient data to show the aim of this study (i.e. an enhanced anti-colon cancer effects on the cells), we prefer using these data without data addition.

Thank you for these valuable suggestions. We give a suggestion in the lines 443-447 and 461-463 in both discussion and conclusion sections.